# A cross-reactive human IgA monoclonal antibody blocks SARS-CoV-2 spike-ACE2 interaction

Monir Ejemel[1,5], Qi Li[1,5], Shurong Hou [2,5], Zachary A. Schiller [1,5], Julia A. Tree[3], Aaron Wallace[1], Alla Amcheslavsky[1], Nese Kurt Yilmaz[2], Karen R. Buttigieg[3], Michael J. Elmore [3], Kerry Godwin[3], Naomi Coombes[3], Jacqueline R. Toomey[1], Ryan Schneider[1], Anudeep S. Ramchetty[1], Brianna J. Close [4], Da-Yuan Chen[4], Hasahn L. Conway [4], Mohsan Saeed[4], Chandrashekar Ganesa[1], Miles W. Carroll[3], Lisa A. Cavacini [1✉], Mark S. Klempner[1✉], Celia A. Schiffer [2✉] & Yang Wang [1✉]

COVID-19 caused by SARS-CoV-2 has become a global pandemic requiring the development of interventions for the prevention or treatment to curtail mortality and morbidity. No vaccine to boost mucosal immunity, or as a therapeutic, has yet been developed to SARS-CoV-2. In this study, we discover and characterize a cross-reactive human IgA monoclonal antibody, MAb362. MAb362 binds to both SARS-CoV and SARS-CoV-2 spike proteins and competitively blocks ACE2 receptor binding, by overlapping the ACE2 structural binding epitope. Furthermore, MAb362 IgA neutralizes both pseudotyped SARS-CoV and SARS-CoV-2 in 293 cells expressing ACE2. When converted to secretory IgA, MAb326 also neutralizes authentic SARS-CoV-2 virus while the IgG isotype shows no neutralization. Our results suggest that SARS-CoV-2 specific IgA antibodies, such as MAb362, may provide effective immunity against SARS-CoV-2 by inducing mucosal immunity within the respiratory system, a potentially critical feature of an effective vaccine.

[1] MassBiologics of the University of Massachusetts Medical School, Boston, MA, USA. [2] Biochemistry and Molecular Pharmacology, University of Massachusetts Medical School, Boston, MA, USA. [3] National Infection Service, Public Health England, Porton Down, Salisbury, Wiltshire, UK. [4] National Emerging Infectious Diseases Laboratories, Boston University, Boston, MA, USA. [5]These authors contributed equally: Monir Ejemel, Qi Li, Shurong Hou, Zachary A. Schiller. ✉email: Lisa.Cavacini@umassmed.edu; Mark.Klempner@umassmed.edu; Celia.Schiffer@umassmed.edu; Yang.Wang@umassmed.edu

In December 2019, a novel coronavirus (SARS-CoV-2) was identified as the cause of an outbreak of acute respiratory infections. The coronavirus disease 2019 (COVID-19) ranges from mild to severe acute respiratory infection, with a fatality rate estimated to range from 2 to 3%[1–4]. Within 3 months of the first report cases, COVID-19 rapidly disseminated through the human population and had become a global pandemic by March 2020. Phylogenic analysis has classified SARS-CoV-2 within the sarbecoviruses subgenus, the β lineage that also contains SARS-CoV, sharing ~79.6% sequence identity[4].

Interventions for the prevention or treatment of COVID-19 are crucial for the ongoing outbreak. Pre- or post-exposure immunotherapies with neutralizing antibodies, would be of great use by providing immediate mucosal immunity against SARS-CoV-2. Although concerns, as occurred with SARS-CoV[5,6], that vaccines may cause disease enhancement still need to be addressed. The feasibility of human monoclonal antibodies (MAbs) as immunoprophylaxis or therapy against coronaviruses including SARS-CoV[7–10] and MERS-CoV[11] has been demonstrated. These anticoronavirus MAbs primarily target the viral spike (S) glycoprotein, a type I transmembrane glycoprotein that produces recognizable crown-like spike structures on the virus surface. The receptor-binding domain (RBD) of the S protein facilitates viral entry into human cells through human angiotensin-converting enzyme 2 (ACE2) receptor binding leveraging a similar mechanism as SARS-CoV[12–14].

Most current anti-SARS-CoV MAbs neutralize virus by binding to epitopes on the spike protein RBD of SARS-CoV[15]. We and others have demonstrated that neutralizing MAbs that block RBD-ACE2 binding could confer potent protection against SARS-CoV as both prophylaxis and treatment in various animal models[7,9,10]. Several anti-SARS-CoV MAbs have demonstrated cross-neutralizing activities against the S protein of SARS-CoV-2[16,17].

Antibody-dependent enhancement of viral infections are one of the major hurdles in the development of effective vaccines. This enhancement is likely facilitated by the Fc domain of IgG but not for its isotype variant IgA[18]. The avidity of mucosal IgA, in comparison with IgG, owing to the multimeric structure, enhances the antibody binding with antigens. In addition, the diverse, high level of glycosylation of IgA antibodies, further protects the mucosal surface with non-specific interference. In animal models, high titers of mucosal IgA in the lung is correlated with reduced pathology upon viral challenge with SARS-CoV[19]. How precisely which isotype may protect the mucosa from SARS-CoV-2 infection remains an open question.

In the current study, we describe the discovery of a cross-neutralizing human IgA monoclonal antibody, MAb362 IgA. This IgA antibody binds to SARS-CoV-2 RBD with high affinity competing at the ACE2 binding interface by blocking interactions with the receptor. MAb362 IgA neutralizes both pseudotyped SARS-CoV and SARS-CoV-2 in 293 cells expressing ACE2. The secretory IgA form of MAb326 also neutralizes authentic SARS-CoV-2 virus. Our results demonstrate that the IgA isotype may play a critical role in SARS-CoV-2 neutralization.

## Results

### Selection of MAb binding to RBD of SARS-CoV-2 in ELISA.
We have previously developed and characterized a panel of human MAbs that targets the RBD of the SARS-CoV S glycoprotein, isolated from transgenic mice expressing human immunoglobulin genes[9,10]. These transgenic mice contains human immunoglobulin genes and inactivated mouse heavy chain and kappa light chain genes (Bristol-Myers Squibb). Transgenic mice were immunized weekly with 10 mg of SARS-CoV spike protein

and adjuvants for 6–8 weeks. Hybridomas were generated following a standard fusion protocol[9]. A panel of over 36 hybridomas were isolated based on various neutralization activities against SARS-CoV with lead antibodies showing protective potency in mice and hamster models[9,10]. To explore the possibility that some of the SARS-CoV-specific hybridoma may have cross-reactivity against SARS-CoV-2, these hybridomas were recovered and screened by ELISA against the SARS-CoV-2 spike protein. MAb362 was identified with cross-binding activity against both the RBD and S1 subunit of the SARS-CoV and SARS-CoV-2 spike proteins (Supplementary Table 1).

While both IgG and IgA are expressed at the mucosa, IgA is more effective on a molar basis and thus the natural choice for mucosal passive immunization as we recently demonstrated in other mucosal infectious disease[20,21]. To further characterize the functionality of MAb362, variable sequences of MAb362 were cloned into expression vectors as either IgG or monomeric IgA isotypes. Both MAb362 IgG and IgA were assessed in ELISA-binding assays against the RBD of the S1 subunit for SARS-CoV ($S_{270–510}$) and SARS-CoV-2 ($S_{319–541}$) (Fig. 1a, b). MAb362 IgA showed better binding activities, compared with its IgG counterpart against SARS-CoV-2 $S_{319–541}$ (Fig. 1b). Assessment of the binding kinetics was consistent with the ELISA-binding trends. The binding affinity of IgA with RBD of SARS-CoV-2 is significantly higher (0.3 nM) than that of IgG (13 nM) due to a much slower dissociation rate as an IgA ($K_{off} = 1.13 \times 10^{-3} \pm 1.06 \times 10^{-4}$) compared with an IgG ($K_{off} = 7.75 \times 10^{-5} \pm 5.46 \times 10^{-5}$) (Fig. 1e, f). Of note, MAb362 IgA and IgG showed similar binding affinity with SARS-CoV $S_{270–510}$ (Fig. 1c, d).

To confirm binding results, the full ectodomain of spike was expressed including residues 1−1208 of SARS-CoV-2 with stabilizing proline mutations and a C-terminal T4 fibritin trimerization motif as described recently[22] (Supplementary Fig. 1). MAb362 IgA still showed better binding activities with the stabilized trimer form as compared with its IgG isotype in ELISA (Fig. 1b) and affinity assays. The binding affinity of MAb362 IgA with the ectodomain of SARS-CoV-2 is 0.17 nM as compared with the 27 nM of IgG (Fig. 1g, h).

### Structural modeling of MAb362 binding to RBD.
To correlate the epitope binding with functionality, MAb362 IgG and IgA were tested in a receptor-blocking assay with Vero E6 cells. The result suggested that both MAb362 IgG and IgA block SARS-CoV-2 RBD binding to receptors in a concentration-dependent manner starting at ~30 nM (Fig. 2a, Supplementary Fig. 2a). Mutational scanning with a combination of alanine (to introduce a loss of interaction), tryptophan (to introduce a steric challenge), and lysine to introduce charge mutations were performed to better delineate the binding surface (Fig. 2b). The results showed that that key residues (Y449A, Y453A, F456A, A475W, Y489A, and Q493W) were critical for the complex and presumably, alterations in the packing caused marked loss of binding affinity (Fig. 2b and Supplementary Fig. 2b). Among the mutant we tested, A475W and Y489A also disrupted ACE2 binding (Supplementary Fig. 3). Interestingly, introduction of lysine mutations had little effect on binding, and some even showed enhanced binding, presumably owing to an overall more favorable charged interaction with the MAb362.

To better define the antibody-binding epitope, known co-crystal and cryo-electron microscopy complexes from SARS-CoV and MERS spike protein in complex with neutralizing antibodies were evaluated for their potential to competitively block ACE2 binding, based on the structural interface of ACE2-SARS-CoV-2-RBD (PDB ID-6VW1)[23]. The 80R-SARS-CoV-RBD complex (PDB ID-2GHW)[24], a crystal structure of SARS-CoV-RBD in

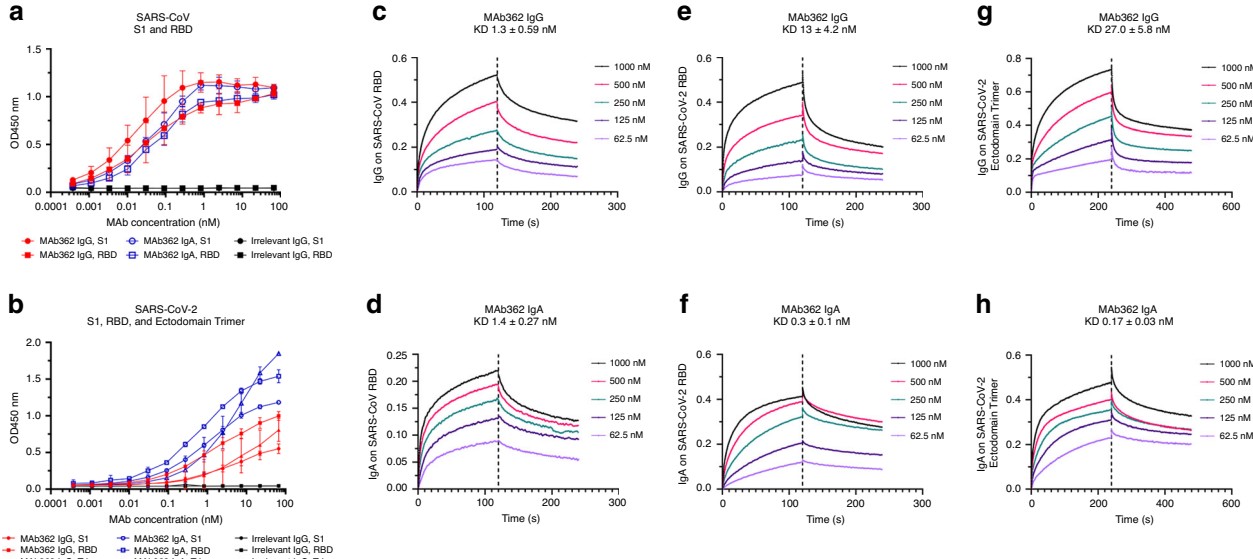

**Fig. 1 Binding of MAb362 IgG and IgA to spikes of SARS-CoV and SARS-CoV-2.** MAb362 IgG and IgA bind to purified SARS-CoV S1 ($S_{1-590}$) and RBD ($S_{270-510}$) truncations **a** and SARS-CoV-2 S1 ($S_{1-604}$), RBD ($S_{319-541}$), and ectodomain trimer **b**. IgGs are red lines, IgAs are blue lines, and irrelevant IgGs are black. Affinity measurements of MAb362 IgG **c**, **e**, **g** and IgA **d**, **f**, **h** against the RBD truncations of S glycoprotein of SARS-CoV and SARS-CoV-2 **c–f**, as well as ectodomain trimer of SARS-CoV-2 **g**, **h** were conducted using bio-layer interferometry and demonstrate nano and sub-nanomolar affinities. Data are plotted as the mean ± s.d. from $n = 3$ independent experiments **a**, **b**. Source data are provided as a Source Data file.

complex with a neutralizing antibody, 80 R, was found most closely to have these characteristics. When the sequence was evaluated, we ascertained that the two antibodies, MAb362 and 80 R, had frameworks with 90% amino-acid sequence identity (Supplementary Fig. 4). Thus, the crystal structure 2GHW provided a suitable scaffold to generate a homology model of MAb362. Protein–protein docking was performed using the Schrodinger suite with tethers based on the mutational analysis. The complex that satisfied the energetics and mutational data was then further interrogated with a 300 ns fully solvated molecular dynamics simulation in which the complex-structure remained stable after equilibration. The final frame of the simulation is the current model of the structure of the MAb362:SARS-CoV-2-RBD complex (Fig. 2c).

The interface of the complex is predicted to form an extensive interface (Fig. 2d and Supplementary Fig. 3) with the CDRs of both the heavy and light chains forming interactions with SARS-CoV-2-RBD. Interestingly, the mutational analysis in combination with this model indicates that the light chain's contribution to this complex may be more significant than the heavy chain (Supplementary Fig. 3c). Complementing the receptor-blocking assay and mutational analysis, our structural analysis further confirms that the MAb362 epitope is directly competing for the ACE2 binding epitope on SARS-CoV-2 spike protein.

**MAb362 structural epitope**. The model of the structure of the MAb362:SARS-CoV-2-RBD complex permitted the superposition of the ACE2:SARS-CoV-2-RBD (PDB: 6VWI[23]) (Fig. 3a). MAb362 is predicted to overlaps with the ACE2 epitope on the RBD. This interface of MAb362 (Fig. 2d) is very similar with the ACE2 interface projected onto the SARS-CoV-2-RBD (Fig. 3b). However, this predicted epitope of MAb362 is different from the other recently reported MAb complexes to the SARS-CoV-2-RBD (Fig. 3c and Supplementary Fig. 5), including: CR3022[17] (PDB: 6W41); S309[16] (PDB: 6WPT); REGN10933 and REGN10987[25]; (PDB: 6XDG); P2B-2F6[26] (PDB: 7BWJ); CB6[27] (PDB: 7C01) and B38[28] (PDB: 7BZ5). MAb362 is predicted to block ACE2-binding interface through a unique epitope conserved between SARS-CoV

and SARS-CoV-2. This finding was consistent with the strong activity of MAb362 of compromising RBD–receptor interaction. As with the binding of ACE2, the predicted MAb362-binding epitope can only be exposed if the RBD was in the open or up conformation in the trimer (Fig. 3d). In the closed conformation, this epitope would not be accessible to MAb362 without major steric clashes. However, unlike CR3022 for instance, MAb362 could access the ACE2-binding epitope(s) if one or more of the trimers is in this open conformation, potentially accounting for the added neutralizing activity.

**MAb362 IgA neutralizes SARS-CoV and SARS-CoV-2**. To evaluate the neutralization potency of cross-reactive MAb362, a pseudovirus assay using lentiviral pseudovirions on 293T cells expressing ACE2 receptor[29] was performed. Both MAb362 IgG and IgA showed potent neutralization activity against SARS-CoV (Fig. 4a). MAb362 IgG weakly neutralized SARS-CoV-2 pseudovirus despite its activities to block receptor binding. Interestingly, isotype switch to MAb362 IgA resulted in significantly enhanced neutralization potency with an $IC_{50}$ value of 1.26 μg ml$^{-1}$, compared with its IgG subclass variant ($IC_{50} = 58.67$ μg ml$^{-1}$) (Fig. 4b). Monomeric MAb362 IgA was also co-expressed with J chain to produce dimeric IgA (dIgA) and secretory component to produce secretory IgA (sIgA) as described in Supplementary Fig. 6[30]. Both dIgA and sIgA were significantly more effective at neutralizing SARS-CoV-2 pseudovirus with an $IC_{50}$ of 30 ng ml$^{-1}$ and 10 ng ml$^{-1}$, respectively (Fig. 4b). Of note, all MAb362 IgG and IgA isotype variants showed comparable neutralization activity against SARS-CoV (Fig. 4a). Further, the most potent form MAb362 sIgA was tested in authentic virus neutralization assay against SARS-CoV-2. MAb362 sIgA neutralized SARS-CoV-2 virus with an $IC_{50}$ value of 9.54 μg ml$^{-1}$ (Fig. 4c). MAb362 IgG failed to neutralize live virus at the highest tested concentration. This is consistent with our prior study showing isotype switch to IgA lead to improved antibody neutralization of HIV infection[31]. Our data extend this observation to coronavirus, suggesting that IgA may play an important role in SARS-CoV-2 neutralization.

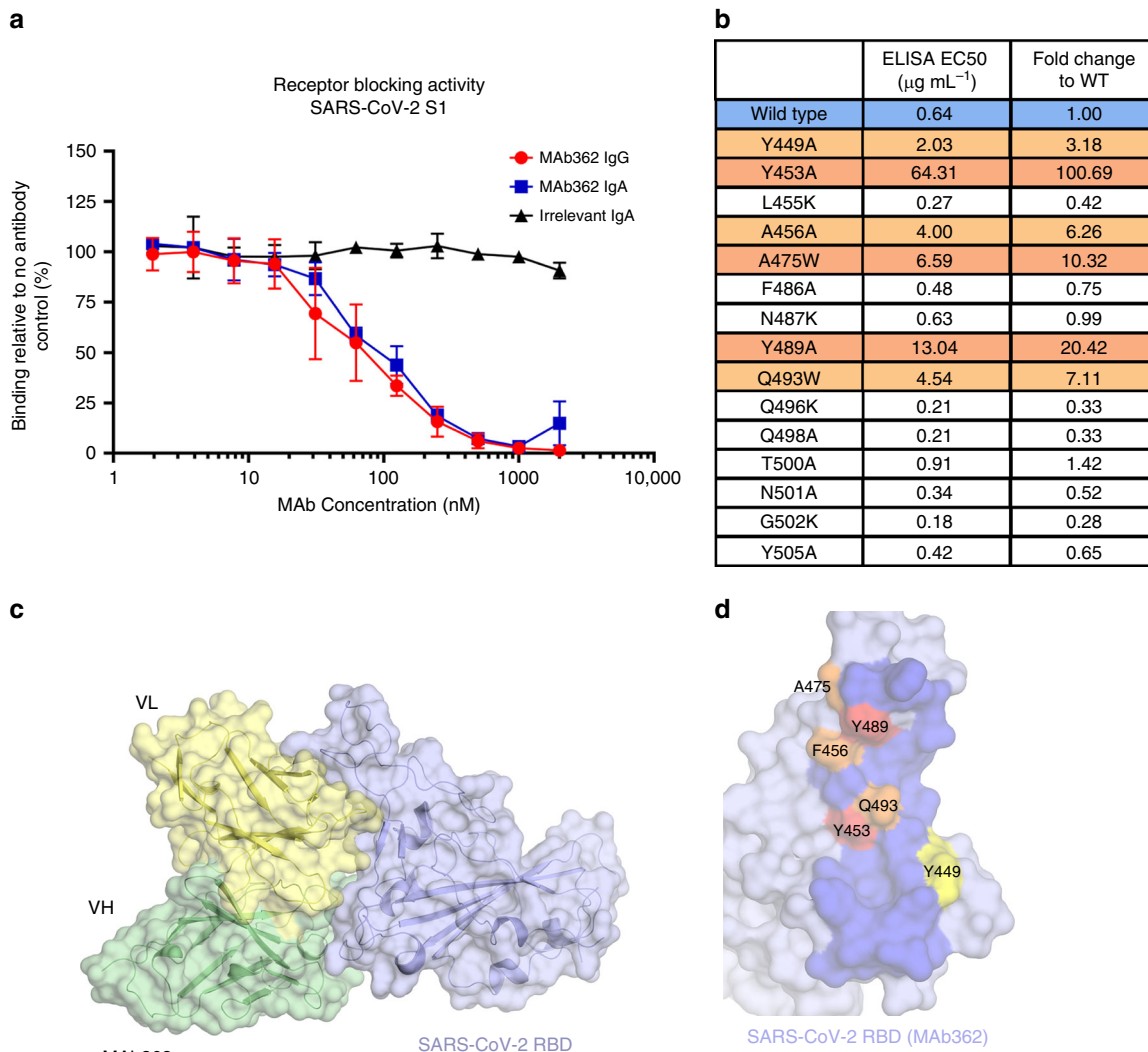

**Fig. 2 Mutationally guided molecular modeling of MAb362 binding to RBD. a** SARS-CoV-2 S1 was pre-incubated with MAb362 IgG (red circles) and IgA (blue squares) ranging from ~2 to 2000 nM. Both MAb362 isotypes demonstrated concentration-dependent inhibition of SARS-CoV-2 RBD binding to Vero E6 cells at concentrations >30 nM. Data are plotted as the mean ± s.d. from $n = 3$ independent experiments. **b** Mutational scanning was performed to better delineate the binding surface. Key residues were mutated and expressed as recombinant proteins. Identified critical residues (orange) were experimental confirmed by shifts in $EC_{50}$ values for MAb362 binding in ELISA relative to wild-type RBD (blue). $EC_{50}$ values calculated from $n = 3$ independent experiments. **c** Surface representation of the predicted molecular model of MAb362 SARS-CoV-2 RBD complex; the light chain of MAb362 (light yellow), the heavy chain (green), and the SARS-CoV-2 RBD (violet). **d** The predicted binding interface on SARS-CoV-2 RBD with MAb362. The residues identified by mutagenesis from **b** are labeled and colored according to influence degree; red represents strongest defects, orange for medium defects and yellow for subtle defects. Source data are provided as a Source Data file **a**, **b**.

## Discussion

This study reports a unique cross-reactive epitope within the core receptor-binding interface of the S protein of both SARS-CoV and SARS-CoV-2. MAb362 IgA neutralizes the virus by competing with S protein binding to ACE2 receptors. Interestingly, our results show that despite the same blocking of spike interaction with ACE2, MAb362 IgG weakly neutralizes SARS-CoV-2, whereas IgA as monomer, dimer, or secretory antibody has significantly enhanced neutralization potency. Structural studies demonstrated that IgA1 has a lengthy hinge region with a 13-a.a. insertion and a relaxed "T" like structure as compared with the more rigid "Y" like structure in IgG[32,33]. Thus, the increase flexibility of IgA1 would likely afford a greater reach toward its epitopes on the target and decrease steric hindrance. MAb362 IgA binds when the spike protein (trimer) is in open form. The longer IgA1 hinge may allow two Fabs to reach two RBDs of the trimer at the same time without clashes, which may not be achieved by the shorter hinge in IgG. Our results suggest that compared with IgG, SARS-CoV-2-specific IgA antibody may play an important independent role in providing protective mucosal immunity. A similar finding has been observed for IgA antibodies to other viruses such as influenza and HIV. When monoclonal antibodies are expressed as IgG or IgA1 isotypes with identical variable regions, antibody binding (affinity, breadth) as well as neutralization are enhanced as IgA1 molecules[34,35]. Though serum half-life of monomeric IgA is relatively short, at the mucosal epithelial interface, IgA is typically present as secretory antibody. Polymeric IgA (predominantly as a dimer) is produced by local plasma cells and binds to the polymeric Ig receptor at the basolateral surface, is transported through the epithelial cell for release at the apical side as secretory IgA, which is the polymeric IgA with addition of the secretory component contributed by the polymeric Ig receptor. The secretory component protects the IgA from harsh conditions prolonging half-life. In addition, the

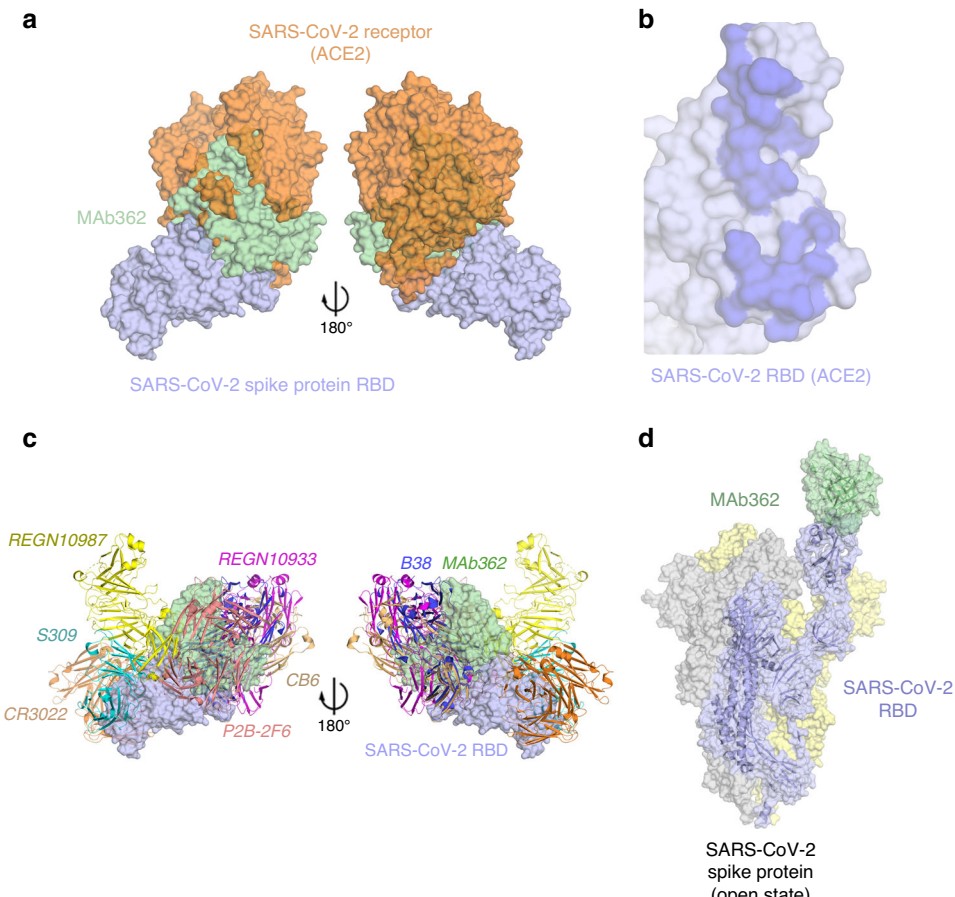

**Fig. 3 Predicted MAb362 structural epitope. a** Superposition of the space filling molecular model of MAb362 (green) complex on the crystal structure of the complex of ACE2 (orange) -SARS-CoV-2 RBD (violet) (6VW1[23]) two views are rotated 180°. **b** The binding interface on SARS-CoV-2 RBD with ACE2 calculated from the co-crystal structure of the complex. The binding interface shown as darker shade is defined as having vdW contacts great than −0.5 kcal mol$^{-1}$. **c** Positioning of MAb362 on SARS-CoV-2 RBD (violet) relative to the binding of other currently published SARS-CoV-2 RBD-neutralizing antibodies: CR3022 (PDB: 6W41[46]; orange); S309 (PDB: 6WPT[16]; cyan); REGN10933 and REGN10987 (PDB: 6XDG[25]; magenta and yellow); P2B-2F6 (PDB: 7BWJ[26]; salmon); CB6 (PDB: 7C01[27]; wheat) and B38 (PDB: 7BZ5[28]; blue). MAb362 recognized a unique epitope overlapping with the binding interface of ACE2. **d** Predicted MAb362 molecular model on the spike trimer in open conformation with one RBD domain exposed 6VYB[45].

carbohydrate moieties of sIgA molecules can bind to adhesion molecules expressed by many pathogens and interfere initial binding of virus to the target cells as the first line of defense. Thus, mucosal passive immunization of secretory IgA directly to the infection site could additionally be an effective approach of systemic delivery of other IgG treatment to achieve immediate protection. To date, innovative approaches are being explored for sIgA production in mammalian and especially plant expression systems for cost-effective production including ongoing work in our laboratories[36,37].

Other recent structure studies have characterized antibodies targeting the RBD domain distal from the receptor-binding core interface of SARS-CoV-2 but lack the characteristics of how MAb362 interacts the ACE2-binding epitope. These neutralizing IgGs, 47D11 and 309, neutralize SARS-CoV-2 with high potency, but do not block receptor binding to ACE[16,38]. Potentially, ACE2 may not be the sole receptor for SARS-CoV-2, similar to SARS-CoV[39], or these antibodies may prevent a conformational change necessary for viral entry. Further study of the interaction between MAb362, and other receptor blocking and neutralizing antibodies against SARS-CoV-2 will provide insight into the design of vaccine and prophylactic/therapeutic antibodies against future emerging infections caused by this viral family.

## Methods

**S glycoprotein expression and purification.** The amino-acid sequence of the SARS-CoV S glycoprotein (Urbani strain, National Center for Biotechnology Information [strain no. AAP13441]) and SARS-CoV-2 S glycoprotein sequence (GeneBank: MN908947) were used to design a codon-optimized version for mammalian cell expression of the gene encoding the ectodomain of the S glycoproteins a.a. 1–1190 ($S_{1-1190}$) for SARS-CoV and a.a. 1–1255 ($S_{1-1255}$) for SARS-CoV-2[22]. The synthetic gene was cloned into pcDNA 3.1 Myc/His in-frame with c-Myc and 6-histidine epitope tags that enabled detection and purification. Truncated soluble S glycoproteins were generated by polymerase chain reaction (PCR) amplification of the desired fragments from the vectors encoding $S_{1255}$ and $S_{1273}$. The SARS-CoV-2 RBD constructs carrying point mutation were generated by following the standard protocol from QuikChange II XL Kit (Agilent). The cloned genes were sequenced to confirm that no errors had accumulated during the PCR process. All constructs were transfected into Expi293 cells using ExpiFectamine 293 Transfection Kit (Thermo Fisher).

The plasmid of stabilized trimer of ectodomain of SARS-CoV-2, NIAID VRC7471, and its expression and purification protocol was kindly provided by Dr. Kizzmekia S. Corbett, PhD, at Vaccine Research Center of National Institute of Allergy and Infectious Diseases as part of large-scale production contract awarded to MassBiologics of UMMS (U24AI126683)[22] In this construct, a gene encoding residues 1−1208 of SARS-CoV-2 S glycoprotein sequence (GenBank: MN908947) was modified by adding two proline substitutions at residues 986 and 987, a "GSAS" substitution at residues 682–685, a C-terminal T4 fibritin trimerization motif, an HRV3C protease cleavage site, a TwinStrepTag and an 8x HisTag. The construct was cloned into the mammalian expression vector pCDNA 3.1. The construct was then transfected into Expi293 cells using ExpiFectamine 293 Transfection Kit (Thermo Fisher). Protein was purified from using StrepTactin resin (IBA) followed by size-exclusion chromatography using a Superose 6 10/300 column (GE Healthcare).

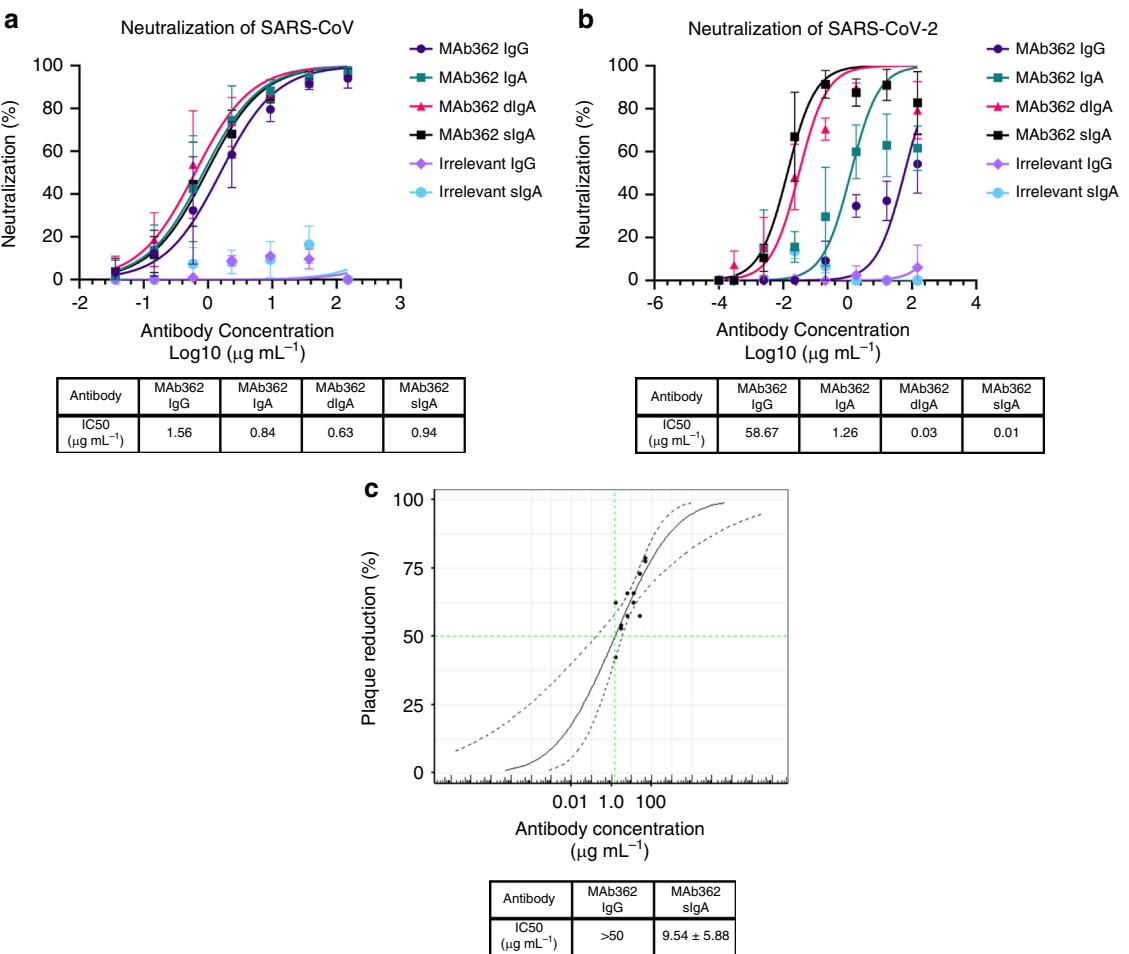

**Fig. 4 IgA isotype switch enhances MAb362 neutralization of SARS-CoV-2.** MAb362 antibody-mediated neutralization of luciferase-encoding pseudovirions with spike proteins of SARS-CoV **a** and SARS-CoV-2 **b**. SARS-CoV and SARS-CoV-2 pseudovirions pre-incubated with serial dilutions of MAb362 were used to infect 293 cells expressing ACE2 receptor. Pseudoviral transduction was measured by luciferase activities in cell lysates 48 h post transduction to calculate neutralization (%) relative to non-antibody-treated controls. IC$_{50}$ values were calculated by nonlinear regression analysis using Prism version 8.1.1. Isotype switching improved SARS-CoV-2 IC50. Data are plotted as the mean ± s.d. from $n = 3$ independent experiments **a**, **b**. **c** Dose–response curve for PRNT with MAb362 at a starting concentration of 50 μg mL$^{-1}$ titrated 1:2. MAb362 sIgA had a 50% endpoint titer of 9.54 ± 5.88 μg mL$^{-1}$ calculated by Spearman–Kärber method, from $n = 2$ biologically independent experiments. Representative data are plotted with a Probit mid-point analysis curve ± 95% CI from one experiment with $n = 2$ technical replicates, using R programming language v3.5.3 and Library ggplot2 v3.3.0 for statistical computing and graphics[47,48]. Source data are provided as a Source Data file **a–c**.

All recombinant proteins were purified by immobilized metal chelate affinity chromatography using nickel-nitrilotriacetic acid (Ni-NTA) agarose beads. Proteins were eluted from the columns using 250 mmol/L imidazole and then dialyzed into phosphate-buffered saline (PBS), pH 7.2 and checked for size and purity by sodium dodecyl sulfate polyacrylamide gel electrophoresis (SDS-PAGE). The stabilized trimer is also analyzed by high performance liquid chromatography (HPLC) (Supplementary Fig. 1).

**Generation of MAbs.** Previously generated frozen hybridomas of anti-SARS-CoV MAbs[9] were recovered and scaled up. Hybridoma supernatants were screened for reactivity to the SARS-CoV-2 S protein. Positive cell clones were selected for antibody sequencing. For MAb362, the heavy chain and light chain variable regions were amplified from hybridoma cells and cloned into an immunoglobulin G1 (IgG) expression vector. Isotype switching was conducted using primers designed to amplify the variable heavy chain of the IgG antibody. Products were digested and ligated into a pcDNA 3.1 vector containing the heavy constant IgA1 chain. The vector was transformed in NEB5-α-competent cells, and sequences were verified ahead of transient transfection. IgG and monomeric IgA1 antibodies were transfected in Expi293 cells. Cell supernatants were harvested 5 days post transfection for antibody purification by protein A sepharose for IgG and Capto L resin for IgA (GE life Sciences). For dimeric IgA1 (dIgA), the heavy and light chain vectors were co-transfected with pcDNA-containing DNA for the connecting J chain. For secretory IgA1 (sIgA) expression, a pcDNA-vector containing gene sequence of secretory component was added to the transfection reaction in a 1:1 ratio. Supernatant was run through a column of Capto L resin to capture the light chain

of antibodies (GE life Sciences). Purified antibodies were dialyzed against PBS before being moved onto size-exclusion chromatography on fast performance liquid chromatography to separate out the desired dimeric or secretory antibodies using a HiLoad 26/600 Superdex 200-pg size-exclusion column (GE life Sciences). The desired fractions were pooled, concentrated, and quality analyzed by SDS-PAGE and HPLC with representative HPLC profile and gel image shown in Supplementary Fig. 6[30].

**ELISA.** Dilutions of purified MAbs were tested in ELISA for reactivity against recombinant S protein. In brief, 96-well plates were coated with S proteins followed by incubation overnight at 4°C. The plates were blocked with 1% BSA with 0.05% Tween 20 in PBS. Hybridoma supernatant or purified antibody diluted in 1× PBS plus 0.1% Tween 20 and added to the 96-well plates and incubated for 1 h at room temperature. Plates were stained with horseradish peroxidase-conjugated anti-kappa (Company Southern biotech, #2060-05,1:2000 dilution) for 1 h and developed using 3,3′,5,5′-tetramethylbenzidine. Absorbance at an optical density at 450 nm (OD450) was measured on an Emax precision plate reader (Molecular Devices) using Softmax Pro v4.3.1 LS.

**ELISA-based ACE2-binding assay.** In all, 250 ng of ACE2 protein was coated on ELISA plates overnight at 4 °C. After blocking with 1% BSA in PBS with 0.05% Tween 20 for 1 h at room temperature, threefold of serial dilutions started from 10 μg ml$^{-1}$ of wild type and point mutations S protein were added into the plates and incubated for 1 h at room temperature. Then plates were stained with mouse-anti-Myc antibody

(BD Pharmingen #551101), at 2 μg ml$^{-1}$ for 1 h, followed by horseradish peroxidase-conjugated goat anti-mouse (Jackson ImmnuoResearch #115-035-062, 1:2000 dilution) for 1 h and developed using 3,3′,5,5′-tetramethylbenzidine. Absorbance at an optical density at 450 nm (OD450) was measured on an Emax precision plate reader (Molecular Devices) using Softmax Pro v4.3.1 LS.

**Flow cytometry-based receptor-binding inhibition assay**. Vero E6 cells were harvested with PBS containing 5 mM ethylenediaminetetraacetic acid and aliquoted to $1 \times 10^6$ cells per reaction. Cells were pelleted then resuspended in PBS containing 10% FBS. Before mixing with the cells, Myc-tagged SARS-CoV-2 $S_{1-604}$ was incubated with the MAb at varying concentrations for 1 h at room temperature, then the S protein was added to the Vero cells to a final concentration of 10 nM. The cells–S protein mixture was incubated for 1 h at room temperature. After incubation, the cell pellets were washed and then resuspended in PBS with 2% FBS and incubated with 10 μg mL$^{-1}$ of mouse-anti-Myc antibody (BD Pharmingen #551101, 1:100 dilution) for 1 h at 4 °C. Pellets were washed again then subsequently incubated with a phycoerythrin-conjugated anti-mouse IgG (Jackson ImmunoResearch, #115-116-071, 1:20 dilution) for 40 min at 4 °C. Cells were washed twice then subjected to flow cytometric analysis using a MACSquant Flow Cytometer (Miltenyi Biotec) and analyzed by MACSQuantify Software v2.11 and FlowJo v10. Binding was expressed as relative to cells incubated with S proteins only.

**Pseudotyped virus neutralization assay**. Pseudovirus was generated employing an HIV backbone that contained a mutation to prevent HIV envelope glycoprotein expression and a luciferase gene to direct luciferase expression in target cells (pNL4-3.Luc.R–E–, obtained from Dr. Nathaniel Landau, NIH). SARS-S and SARS2-S spike protein was provided in trans by co-transfection of 293 T cells with pcDNA-G with pNL4-3.Luc.R–E–. Supernatant containing virus particles was harvested 48–72 h post transfection, concentrated using Centricon 70 concentrators, aliquoted, and stored frozen at −80 degree. Before assessing antibody neutralization, the 293 T cells were transient transfected with 100 ng pcDNA-ACE2 each well in 96-well plates, and the cells were used for the pseudovirus infection 24 hs after transfection. A titration of pseudovirus was performed on 293 T cells transiently transfected with human ACE2 receptor to determine the volume of virus need to generate 50,000 counts per second (cps) in the infection assay. The appropriate volume of pseudovirus was pre-incubated with varying concentrations of MAbs for 1 h at room temperature before adding to 293 T cells expressing ACE2. 24 h after the infection, the pseudovirus was replaced by the fresh complete media, and 24 h after media changing the infection was quantified by luciferase detection with BrightGlo luciferase assay (Promega) and read in a Victor3 plate reader (Perkin Elmer) for light production.

**Plaque reduction neutralization assay (PRNT)**. Monoclonal antibody was serially diluted and incubated with ~70 plaque forming units of wild-type SARS-CoV-2 (2019-nCoV/Victoria/1/2020), for 1 h at 37 °C in a humidified box. The virus/antibody mixture was then allowed to absorb onto monolayers of Vero E6 [(ECACC 85020206, European Collection of Authenticated Cell Cultures, UK] for 1 h at 37 °C in a humidified box. Overlay media [MEM (Life Technologies, California, USA) containing 1.5% carboxymethylcellulose (Sigma), 5% (v/v) fetal calf serum (Life Technologies) and 25 mM 4-(2-hydroxyethyl)-1-piperazineethanesulfonic acid buffer (Sigma)] was added and the 24-well plates were incubated in a humidified box at 37 °C for 5 days. Plates were fixed overnight with 20% (w/v) formalin/PBS, washed with tap water and stained with methyl crystal violet solution (0.2% v/v) (Sigma). The neutralizing antibody titers were defined as the amount of antibody (μg mL$^{-1}$) resulting in a 50% reduction relative to the total number of plaques counted without antibody, by performing a Spearman–Kärber analysis[40] using Microsoft Excel v2016. An internal positive control for the PRNT assay was run using a sample of human MERS convalescent serum known to neutralize SARS-CoV-2 (National Institute for Biological Standards and Control, United Kingdom).

**Structural modeling analyses**. Three crystal structures, 2GHW the complex of 80 R:SARS-CoV-RBD[24], 2AJF the complex of ACE2:SARS-CoV-RBD[41], and 6VW1 the complex of ACE2:SARS-CoV-2-RBD[23] were used as initial scaffolds in the determinations of the models of MAb362:SARS-CoV-RBD and MAb362:SARS-CoV-2-RBD. The amino-acid sequence of MAb362 was aligned to the amino-acid sequences of 80 R bound SARS-CoV-1 crystal structure (PDB: 2GHW). The point mutational studies of SARS-CoV-2 RBD were used as restraints to guide the protein–protein docking of MAb362 against SARS-CoV-2 RBD. The docking was performed using Glide (Schrödinger software suite v19-4) and Modeller v9.23. The highest scored docking pose that also best satisfied the mutational analysis was further optimized through 300 ns molecular dynamic (MD) simulations. The MD simulations were performed using Desmond (Schrödinger software suite v19-4)[42–44]. The final frame of the MD simulations was used as the final structural model of MAb362-RBD complex.

The structural model of MAb362 binding to the SARS-CoV-2 spike trimer was based on 6VYB[45]. All figures were made within PyMOL Molecular Graphics System v2.3.4 (Schrödinger). The residue van der Waals potential between the various complexes was extracted from the structures energies using the energy potential within Desmond.

**Mutational scanning to identify MAb362-binding residues**. SARS-CoV-2 RBD residues were individually mutated with a combination of alanine (to introduce a loss of interaction), tryptophan (to introduce a steric challenge), and lysine mutations to introduce charge using QuikChange II XL Kit (Agilent) or BioXp 3200 System (SGI-DNA). The genes were cloned into RBD expression vectors and RBD proteins were purified as described above. Mutant RBDs were confirmed intact expression on proteins gels, and the same amount of proteins were coated on the plate for ELISA assays.

Dilutions of purified MAbs were tested in ELISA for reactivity against mutant RBD proteins. In all, 96-well plates were coated with 100 μl of 5 μg of RBD mutants followed by incubation overnight at 4 °C. The plates were blocked with 1% BSA with 0.05% Tween 20 in PBS. Purified antibody diluted in 1× PBS plus 0.1% Tween 20 and added to the 96-well plates and incubated for 1 h at room temperature. Plates were stained with alkaline phosphatase affiniPure goat anti-Human IgG (Jackson ImmunoResearch #109-055-098, 1:1000 dilution) for 1 h at room temperature. Alkaline phosphatase affiniPure goat anti-Mouse IgG (Jackson ImmunoResearch #115-055-003, 1:1000 dilution) was used to detect his tag in a separate ELISA to verify protein expression and coating. Plates were developed using p-Nitrophenyl Phosphate (Thermo Fisher Scientific). Absorbance at an optical density at 405 nm (OD405) was measured on an Emax precision plate reader (Molecular Devices) using Softmax Pro v4.3.1 LS. ELISAs assay was performed to determine binding of the MAbs to the mutant proteins compared with the wild type. Key residues were identified by RBD mutations that reduced EC$_{50}$ values relative to the wild-type RBD.

**Affinity determination for MAb362**. Bio-layer interferometry (BLI) with an Octet HTX (PALL/ForteBio) was used to determine the affinity of MAb362 IgG and IgA1 to the RBD of SARS-CoV and SARS-CoV-2 S protein. MAbs were added to 96 wells plates at 1000 nM and titrated 1:2 to 62 nM using PBS. RBD of SARS-CoV, RBD, and ectodomains of SARS-CoV-2 were biotinylated (Thermo Fisher) and immobilized on Streptavidin Biosensors (ForteBio) for 120 s at 1600 nM concentration. After a baseline step, MAb362-antigen binding rate was determined when the biosensors with immobilized antigen were exposed to MAb362 IgG or IgA1 at different concentrations for 120 s. Following association, the MAb362-RBD complex was exposed to PBS and the rate of the MAb362 dissociation from antigen was measured. Each assay was performed in triplicate. Binding affinities for MAb362 were calculated using association and dissociation rates with ForteBio Data analysis software v8.1 (PALL).

**Statistical analysis**. Statistical calculations were performed using Prism version 8.1.1 (GraphPad Software, La Jolla, CA). EC$_{50}$ and IC$_{50}$ values were calculated by sigmoidal curve fitting using nonlinear regression analysis.

**Reporting summary**. Further information on research design is available in the Nature Research Reporting Summary linked to this article.

## Data availability

Antibody heavy and light chain sequence can be found on GenBank (MAb362 Heavy Chain Accession # MT789771, MAb362 Light Chain Accession # MT789772). Database files used in the study include: PDB 2GHW, the complex of 80 R:SARS-CoV-RBD24; PDB 2AJF, the complex of ACE2:SARS-CoV-RBD43 and PDB 6VW1, the complex of ACE2:SARS-CoV-2-RBD. All other data generated are included in figures and tables in this published article. Source data are provided with this paper.

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

## Acknowledgements

We thank Dr. Robert Finberg at UMMS for helpful advice on functional assays. The initial mouse immunization work in 2005 was supported by NIH Contract NO1-AI-65315. S.H., N.Y.K., and C.A.S. were supported by NIH R01AI150478. The production of stabilized ectodomain trimer was supported by NIH contract U24AI126683. The dIgA and sIgA expression platform was supported by Bill & Melinda Gates Foundation.

## Author contributions

M.E., Q.L, A.W. cloned, expressed, and purified MAbs, S proteins, truncations, and variants with assistance from A.A., J.R.T., A.S.R., R.S., and C.G.; M.E. and Q.L. designed and performed affinity and binding assays and flow cytometry with assistance from Y.W., L.A.C., and Z.A.S; S.H., carried out structural modeling and analyses with assistance from Q.L. N.Y.K., Y.W., and C.A.S.; M.E. and Q.L. conducted pseudovirus neutralization with assistance from B.J.C., D.-Y.C., H.L.C., S.M., L.A.C., and Y.W.; J.A.T, K.R.B., M.J.E., K.G., N.C., and M.W.C. designed and conducted live virus neutralization and analysis; Z.A.S. and J.A.T. conducted data and statistical analysis with assistance from M.E., Q.L, and Y.W.; Z.A.S and Y.W. wrote the paper with assistance from M.E., Q.L., L.A.C., M.S.K., J.A.T., and C.A.S.; M.S.K., L.A.C., and Y.W. supervised the project.

## Competing interests

A patent application has been filed on 5 May 2020 on monoclonal antibodies targeting SARS-CoV-2 (U.S. Patent and Trademark Office patent application no. 63/020,483; patent applicants: Y.W., M.E., Q.L., and M.K., University of Massachusetts Medical School). The remaining authors declare no competing interests.
