## [Peer Review File · Nature Communications]

REVIEWER COMMENTS

Reviewer #1 (Remarks to the Author):

This manuscript describes a SARS/SARS2 cross-reactive antibody that was initially generated in a humanized mouse model a number of years ago during a SARS study. The authors show that the antibody likely binds to the ACE2 receptor binding site and blocks receptor engagement. The authors make the antibody as an IgG and IgA and observe better binding and neutralization in the IgA format, which they then suggest may have better in vivo properties for combatting SARS2 due to the preferential deposition in the mucosal tissue. The study is of moderate interest but there are a number of glaring and egregious mistakes and disconnected conclusions that give one serious pause as to the quality of the study. First and foremost is the “neutralization” assay that is used and the fact that they do not even report an IC50 value for comparison to other RBD antibodies.

Major concerns:

Fig. 1 the dynamic range of the association step is very small and it looks like the curves are appear not to be normalized to 0.

It is difficult to understand why the IgA binds better than IgG to SARS-CoV-2 by Octet. Is it the dimeric version? If not then it should have the same variable regions and therefor bind similarly just like SARS-CoV.

It would be good to show a sequence alignment between 80R and mAb362, both heavy and light chains. I assume only the heavy chain is 90% identical in framework. Also, are they predicted to be derived from the same germline genes?

Additionally, just because the framework was 90% identical does not mean that the binding is similar in any way as the paratope is formed by the much more variable CDR loops.

There are better programs to model antibody structures than Modeller, namely RosettaAntibody and RosettaAntibodyDesign. Also, given the level of modeling, prediction of specific contacts down to the level of Van der Waals and hydrogen bond is meaningless.

While the mutagenesis studies suggest that they are in the right region of the RBD the modeling and docking based on 8OR is not guaranteed to be very accurate. A crystal structure of the Fab alone or even better the Fab-RBD complex would be a significant addition to this paper.

In the first paragraph of the results it would be nice to provide a short description of the immunization or infection and mouse model that was used to generate the panel of anti-SARS CoV mAbs that includes mAb362. Otherwise there is not appropriate context for the source of these antibodies even though the original study was properly referenced.

IgG dimers are notoriously difficult to produce and are often heterogenous from monomers to oligomers when express with J chain. It is therefore important to run size exclusion chromatography to purify just the dimeric fraction. Otherwise there is likely a mixture of valencies, which confounds any binding studies.

The ability of monomeric IgA to more potently neutralize than IgG is still somewhat controversial and there is only one other example in the HIV literature. Thus, it would have been nice if the authors made other SARS or SARS-CoV-2 control antibodies in both formats and tested binding and neutralization. I also find it curious that the authors did not test (or show) the dimeric IgA neutralization data for SARS in Fig. 4a.

The emerging literature suggests that pseudovirus neutralization is a less good measure than live virus neutralization so the authors should consider redoing the neutralization assays.

IC50 values for neutralization were either not calculated or not included for some reason, although the neutralization curves were shown in Fig. 4. Further, the abstract states that mAb362 neutralizes human epithelial cells expressing ACE2, which is very misleading. The authors instead used HEK 293 cells transfected with ACE2 receptor. The E in HEK stands for "embryonic" not "epithelial". Further, HEK cells were generated via a Adeno infection of an unknown cell type and are quite strange cells. Thus, while the ACE2 transfection is probably sufficient to measure blocking of infection by an antibody it is hardly representative of lung epithelium as suggested by the abstract. I am therefore lead to believe that this was an honest mistake out of ignorance or carefully crafted language to increase the impact of the study.

Reviewer #2 (Remarks to the Author):

The manuscript by Wang et al describes the discovery and characterisation human IgA monoclonal antibody, MAb363, which is cross-reactive for SARS-CoV and SARS-CoV-2. They demonstrate that Mab362 binds to the spike glycoprotein and competitively blocks the interaction with the human ACE2 receptor. Furthermore, they show that Mab362 can neutralise pseudotyped SARS-CoV and SARS-CoV-2 in human epithelial cells expressing hACE2, suggesting that IgA may play an important role in SARS-CoV-2 neutralization. However, the results, although important, lack robustness. Below are a number of experiments which could sufficiently strengthen the conclusions to warrant publication.

Major concerns

1) It is stated in the materials and methods section (lines 384-389) that recombinant spike ectodomains were produced, but the reported ELISA and bio-layer interferometry experiment were performed on the RBD only. Given that the proposed Mab363 epitope is only accessible in the open conformation of spike, binding experiments with RBD alone represent the most favourable, and less physiologically relevant, situation. Indeed, the SARS-CoV-2 spike protein is reported to sample the open conformation even less frequently than SARS-CoV (<https://www.pnas.org/content/early/2020/05/05/2003138117>). Therefore, additional binding experiments should be performed on the ectodomains (which were already produced by the authors). The authors should carefully check the quality of such ectodomains by negative-stain electron microscopy prior to using them for functional experiments to ensure that they display the expected prefusion conformation.

2) No cryo-EM or crystallography data is reported but the authors go into great detail interpreting a predicted model of the Mab362-RBD complex (even going as far as to speculate about a water mediated interaction). Given the potential importance of SARS-CoV-2 neutralising antibodies, it does not seem appropriate to draw such firm conclusions about the binding interface without determining the structure experimentally. This entire section needs to be toned down. While it is not realistic to request that the authors obtain a structure, their predicted epitope could be further verified by introducing targeted SARS-CoV RBD residues into that of SARS-CoV-2 and demonstrating that this leads to increased binding of Mab362. If possible, it would also be nice to see an alignment of the Mab362 and 80R sequences, rather than taking the authors word for it. In addition, I would like to see the authors homology model so that I can compare it to the 80R-RBD complex (PDB-ID 2GHW).

3) Mab362 IgG is weakly neutralising but the IgA1, and its dimeric form, are able to potentially neutralise pseudotype virus. The authors speculate that the increased flexibility of IgA may allow for multivalent interactions, hence explaining its increased potency. The dimeric form shows even greater neutralisation capacity, which I suspect may be due to crosslinking of virions. Given that increased avidity seems to be crucial for Mab362, it would be prudent to perform neutralisation experiments on SARS-CoV-2. The reason being that the pseudotype virus may not faithfully recapitulate the density of spikes which would be encountered on the native virion.

4) I agree that IgA present an interesting therapeutic potential for mucosal protection against viruses. However, in contrast to monoclonal antibodies of the IgG isotype, their development as research tools or human therapeutics has been scarce. Given that the IgG Mab362 show relatively low potency, how feasible is it that the IgA isotype can be immediately translated into a prophylactic or treatment option? It is well known that the half-life of IgA is lower than that of IgG, but the authors do not discuss this.

Minor points

Line 86: Missing space "IgGor IgA1 isotypes".

Line 104: Remove the word "striking".

Line 105: Replace the word "outstanding" with "suitable".

Line 106: Please remove "build a highly accurate atomic". You don't have experimentally derived electron density or Coulomb potential so you had nothing to "build" into. "Highly accurate" implies that you verified your model with an actual structure. It would be sufficient to say "generate a homology model".

Line 110-131: The very detailed interpretation of the proposed model may be misleading to the untrained eye. Wherever possible, it needs to be made explicitly clear that this model is not derived experimentally.

Line 132-133: As discussed above, gain of function would be more convincing.

The labels in figure 2C-E and 2G-H are too small.

Point-by-Point Response to the 05/22/20 Critique

Reviewer #1

We would like to thank the reviewer for a thorough and very helpful evaluation of our manuscript. We sincerely appreciate all valuable comments and suggestions. We have made substantial revision of the manuscript in line with these comments. All the figures and tables have been superseded with newer data. The revisions greatly improved the manuscript. We list below the revisions and new experimental results implemented to address the reviewer's concerns.

Comment 1 *First and foremost is the “neutralization” assay that is used and the fact that they do not even report an IC50 value for comparison to other RBD antibodies.*

Response

As detailed below, the IC₅₀ values from neutralization assays (with both pseudovirus and live virus) are now tabulated in Figure 4 together with the curves used to determine these values.

Comment 2 *Fig. 1 the dynamic range of the association step is very small and it looks like the curves are appear not to be normalized to 0.*

Response

We have normalized the curves to 0, and replaced affinity panels in Figure 1 with new graphs.

Comment 3 *It is difficult to understand why the IgA binds better than IgG to SARS-CoV-2 by Octet. Is it the dimeric version? If not then it should have the same variable regions and therefor bind similarly just like SARS-CoV.*

Response

The IgA tested in original Figure 1 is the monomeric IgA. Different Fc region can result in size or conformational difference in the paratopes, which may affect the binding affinity especially the k_{off} rate as observed in affinity assays. We added discussion of prior published work in this area where monoclonal antibodies with identical variable regions but expressed as both IgG and IgA1 had different binding and functional activity. We also confirmed this binding activity difference using three forms of spike including RBD, S1, and trimerized full-length Spike in ELISA and affinity assay with an updated Figure 1.

Comment 4 *It would be good to show a sequence alignment between 80R and mAb362, both heavy and light chains. I assume only the heavy chain is 90% identical in framework. Also, are they predicted to be derived from the same germline genes? Additionally, just because the framework was 90% identical does not mean that the binding is similar in any way as the paratope is formed by the much more variable CDR loops.*

Response

Both 80R and mAb 362 are 3-30 germline gene. An alignment between 80R and mAb362 framework region is now included as Extended Data Figure 2.

Comment 5 There are better programs to model antibody structures than Modeller, namely RosettaAntibody and RosettaAntibodyDesign. Also, given the level of modeling, prediction of specific contacts down to the level of Van der Waals and hydrogen bond is meaningless.

Response

We have carefully re-evaluated our modeling, with 5 different potential complexes based on various co-crystal structures and molecular models, followed by 300 ns molecular dynamics simulations. The model chosen is the most stable and consistent with the experimental mutational results.

We have also toned down the specifics of the molecular model, removed the in-depth van der Waals analysis, and instead display overall how we expect the two proteins to interact, highlighting the mutational hotspots.

Comment 6 While the mutagenesis studies suggest that they are in the right region of the RBD the modeling and docking based on 80R is not guaranteed to be very accurate. A crystal structure of the Fab alone or even better the Fab-RBD complex would be a significant addition to this paper.

Response

We agree with the reviewer that the model does not replace a co-crystal structure (which we have been actively pursuing); however, the combination of the high homology and the mutational scanning validate the reasonableness of the molecular model. Nevertheless, as accuracy is not guaranteed we toned down the interpretation and level of detail in analysis of the model.

Comment 7 In the first paragraph of the results it would be nice to provide a short description of the immunization or infection and mouse model that was used to generate the panel of anti-SARS CoV mAbs that includes mAb362. Otherwise there is not appropriate context for the source of these antibodies even though the original study was properly referenced.

Response

The first paragraph of the results section was revised to include a short description of immunization of humanized transgenic mice, hybridoma generation and clone selection to generate this panel of mAbs.

Comment 8 IgA dimers are notoriously difficult to produce and are often heterogenous from monomers to oligomers when express with J chain. It is therefore important to run size exclusion chromatography to purify just the dimeric fraction. Otherwise there is likely a mixture of valencies, which confounds any binding studies.

Response

Our purification process for dimeric and secretory IgA antibodies includes size exclusion chromatography. We added an Extended Data Figure 4 with gels and HPLC profiles. We also revised the method section with more detailed description of the purification method.

Comment 9 *The ability of monomeric IgA to more potently neutralize than IgG is still somewhat controversial and there is only one other example in the HIV literature. Thus, it would have been nice if the authors made other SARS or SARS-CoV-2 control antibodies in both formats and tested binding and neutralization. I also find it curious that the authors did not test (or show) the dimeric IgA neutralization data for SARS in Fig. 4a.*

Response

Both dimeric IgA and secretory IgA forms of 362 were tested with SARS-CoV pseudovirus as shown in new panel A of Figure 4. Interestingly, difference in the neutralization potency between IgG and IgA was only observed with SARS-CoV-2 but not with SARS-CoV. This is consistent with the observation in ELISA binding assay. MAb362 is the only mAb we discovered so far with binding and neutralizing activities against SARS-CoV-2. We will continue to test the isotype impact when other groups disclose the sequences of other SARS-CoV-2 neutralizing antibodies in the literatures.

Comment 10 *The emerging literature suggests that pseudovirus neutralization is a less good measure than live virus neutralization so the authors should consider redoing the neutralization assays.*

Response

We agree with the reviewer. We have now tested MAb362 isotype variants including IgG and sIgA in live virus assay to confirm the pseudovirus findings. Our collaborators at National Infection Service of Public Health England at Porton Down, UK performed the assay with two independent experimental repeats. The results are included as a new panel C in Figure 5 and IC₅₀ values described in the text.

Comment 11 *IC50 values for neutralization were either not calculated or not included for some reason, although the neutralization curves were shown in Fig. 4.*

Response

IC₅₀ values for each sample were added as a table below panels of Figure 4.

Comment 12 *Further, the abstract states that mAb362 neutralizes human epithelial cells expressing ACE2, which is very misleading. The authors instead used HEK 293 cells transfected with ACE2 receptor. The E in HEK stands for “embryonic” not “epithelial”. Further, HEK cells were generated via a Adeno infection of an unknown cell type and are quite strange cells. Thus, while the ACE2 transfection is probably sufficient to measure blocking of infection by an antibody it is hardly representative of lung epithelium as suggested by the abstract. I am therefore lead to believe that this was an honest mistake out of ignorance or carefully crafted language to increase the impact of the study.*

Response

We apologize for this misleading description. We have revised the description.

Reviewer #2

We would like to thank the reviewer for a thorough and very helpful evaluation of our manuscript. We sincerely appreciate all valuable comments and suggestions. We have made substantial revision of the manuscript in line with these comments. All the figures and tables have been superseded with newer data. The revisions greatly improved the manuscript. We list below the revisions and new experimental results implemented to address the reviewer's concerns.

The manuscript by Wang et al describes the discovery and characterisation human IgA monoclonal antibody, MAb363, which is cross-reactive for SARS-CoV and SARS-CoV-2. They demonstrate that Mab362 binds to the spike glycoprotein and competitively blocks the interaction with the human ACE2 receptor. Furthermore, they show that Mab362 can neutralise pseudotyped SARS-CoV and SARS-CoV-2 in human epithelial cells expressing hACE2, suggesting that IgA may play an important role in SARS-CoV-2 neutralization. However, the results, although important, lack robustness. Below are a number of experiments which could sufficiently strengthen the conclusions to warrant publication.

Comment 1 *It is stated in the materials and methods section (lines 384-389) that recombinant spike ectodomains were produced, but the reported ELISA and bio-layer interferometry experiment were performed on the RBD only. Given that the proposed Mab363 epitope is only accessible in the open conformation of spike, binding experiments with RBD alone represent the most favorable, and less physiologically relevant, situation. Indeed, the SARS-CoV-2 spike protein is reported to sample the open conformation even less frequently than SARS-CoV (<https://www.pnas.org/content/early/2020/05/05/2003138117>). Therefore, additional binding experiments should be performed on the ectodomains (which were already produced by the authors). The authors should carefully check the quality of such ectodomains by negative-stain electron microscopy prior to using them for functional experiments to ensure that they display the expected prefusion conformation.*

Response

We have now included new ELISA results against S1, and the full-length ectodomains in Figure 1b. The full-length ectodomain was produced as a stabilized trimer. The expression plasmid was kindly provided by Dr. Kizzmekia S. Corbett, PhD, at Vaccine Research Center of National Institute of Allergy and Infectious Diseases as part of large scale GLP production contract awarded to our laboratories (U24AI126683). The detailed protocol was described in Dr. Corbett's recent publication (Wrapp, D. *et al.* Cryo-EM structure of the 2019-nCoV spike in the prefusion conformation. *Science* **367**, 1260-1263, 2020). The trimer was produced by Process Development engineers at MassBiologics (authorship added), and the HPLC profiles of produced proteins are included in Extended Data Figure 1.

Comment 2 *No cryo-EM or crystallography data is reported but the authors go into great detail interpreting a predicted model of the Mab362-RBD complex (even going as far as to speculate about a water mediated interaction). Given the potential importance of SARS-CoV-2 neutralizing antibodies, it does not seem appropriate to draw such firm conclusions about the binding interface without determining the structure experimentally. This entire section needs to be toned down. While it is not realistic to request that the authors obtain a structure, their predicted epitope could be further verified by introducing targeted SARS-CoV RBD residues into that of SARS-CoV-2 and demonstrating that this leads to increased binding of Mab362. If possible, it would also be nice to see an alignment of the*

Mab362 and 80R sequences, rather than taking the authors word for it. In addition, I would like to see the authors homology model so that I can compare it to the 80R-RBD complex (PDB-ID 2GHW).

Response

As explained above in response to Reviewer #1, we have greatly toned down the analysis and re-evaluated the molecule modeling. We did not introduce targeted SARS-CoV RBD residues into SARS-CoV-2 but have performed and included more extensive mutational analysis of the SARS-CoV-2 RBD. The antibody showed neutralization activity is the 362 IgA variant, which already has good binding affinity with SARS-CoV-2 RBD at 0.3 nM affinity due to a much slower dissociation rate compare to its IgG variant. Our new mutational analysis uncovered several mutations that exhibited gain of function activities in Figure 2F.

In addition, an alignment between 80R and mAb362 framework region is now included as Extended Data Figure 2. We also submitted homology model in PDB file as well as the validation report for review, but the databank is not accepting homology model depositions.

Comment 3 *Mab362 IgG is weakly neutralizing but the IgA1, and its dimeric form, are able to potently neutralize pseudotype virus. The authors speculate that the increased flexibility of IgA may allow for multivalent interactions, hence explaining its increased potency. The dimeric form shows even greater neutralization capacity, which I suspect may be due to crosslinking of virions. Given that increased avidity seems to be crucial for Mab362, it would be prudent to perform neutralization experiments on SARS-CoV-2. The reason being that the pseudotype virus may not faithfully recapitulate the density of spikes which would be encountered on the native virion.*

Response

We agree with the reviewer. We have now tested MAb362 isotype variants including IgG and sIgA in live virus assay to confirm the pseudovirus findings. Our collaborators at National Infection Service of Public Health England at Porton Down, UK performed the assay with two independent experimental repeats. The results are included as a new panel C in Figure 5 and IC₅₀ values described in the text.

Comment 4 *I agree that IgA present an interesting therapeutic potential for mucosal protection against viruses. However, in contrast to monoclonal antibodies of the IgG isotype, their development as research tools or human therapeutics has been scarce. Given that the IgG Mab362 show relatively low potency, how feasible is it that the IgA isotype can be immediately translated into a prophylactic or treatment option? It is well known that the half-life of IgA is lower than that of IgG, but the authors do not discuss this.*

Response

Most measures of IgA half-life are based on serum IgA, which is predominantly monomeric. It is not clear if this is the same case for the half-life of sIgA in the lung mucosal epithelia. We have added discussion about the half-life as well as the feasibility of using sIgA as passive immunization directly to the mucosa. Our team is currently working with Bill & Melinda Gates Foundation to establish a sIgA expression platform in mammalian cells as well as large-scale production in plants for clinical trials (unpublished data). We hope this can facilitate the immediate translation of MAb362 sIgA as a treatment option through mucosal administration in combination with other systemically administrated therapeutics.

Minor points

Line 86: Missing space “IgG or IgA1 isotypes”

Line 86 was revised to read “IgG or IgA1 isotypes”

Line 104: Remove the word “striking”.

The word “striking” was removed from Line 104

Line 105: Replace the word “outstanding” with “suitable”.

The word “outstanding” was replaced with “suitable” on Line 105

Line 106: Please remove “build a highly accurate atomic”. You don’t have experimentally derived electron density or Coulomb potential so you had nothing to “build” into. “Highly accurate” implies that you verified your model with an actual structure. It would be sufficient to say “generate a homology model”.

Line 106 was revised to “generate a homology model.”

Line 110-131: The very detailed interpretation of the proposed model may be misleading to the untrained eye. Wherever possible, it needs to be made explicitly clear that this model is not derived experimentally.

As described above, we have toned down the detailed analysis of the interface.

Line 132-133: As discussed above, gain of function would be more convincing.

Our mutational analysis uncovered several mutations that exhibited gain of function activities in Figure 2F.

The labels in figure 2C-E and 2G-H are too small.

Bigger labels were used for new figure 2

REVIEWERS' COMMENTS:

Reviewer #1 (Remarks to the Author):

The authors present a much improved version of the manuscript and have adequately addressed most of the reviewer comments. There are still a few outstanding issues to address.

1. The words "providing mucosal immunity" should be removed from the title as they are misleading. One would need to conduct animal studies to show mucosal immunity.
2. A sequence alignment of the CDRs should also be included in the supplement, not just the framework regions.
3. Even with the improved structural modeling the atomic representations in Figs. 2 and 3 of the mAb362 bound to RBD should be relegated to the supplement to de-emphasize the confidence in the model. The new mutagenesis data is highly supportive of the antibody overlapping the ACE2 binding site but the accuracy of the model really needs to be validated with an explicit structure.

Reviewer #2 (Remarks to the Author):

The authors have made a substantial effort to address all of the points raised previously, and the resulting revised manuscript is greatly improved. In particular, the addition of live virus neutralisation experiments has strengthened their conclusions. I am mostly satisfied with the responses to my specific concerns (many of which were shared by reviewer 1). I would urge the authors to take more care when drafting rebuttal letters in future as they refer to the wrong supplementary figures on several occasions.

The construct and purification strategy used to obtain the spike ectodomains is appropriate and I am satisfied by the ELISA results. However, for the sake of being thorough, I'd like to point out that proteins containing multimerization domains, such as T4 fibritin, can often elute at the expected molecular weight by SEC even if they are not adopting the desired 'intact' quaternary structure.

Given the degree of sequence similarity to 80R, and the mutagenesis data, I'd say the authors homology model is in the right ballpark. By toning down their interpretation of the epitope, the results are less overstated. That being said, X-ray crystallography and/or cryo-EM data would

significantly improve the paper, but I appreciate this is outside the scope of a reasonable reviewer request.

Overall, given the important nature of the work, I believe the manuscript is sufficiently improved to warrant publication.

Point by point response to reviewers' comments:

We would like to thank the reviewers for a thorough and very helpful evaluation of our manuscript. The revisions greatly improved the manuscript. Thank you so much for your support for the acceptance! We list below the revisions and responses to the comments.

Reviewer #1 (Remarks to the Author)

The authors present a much improved version of the manuscript and have adequately addressed most of the reviewer comments. There are still a few outstanding issues to address.

Comment 1. *The words "providing mucosal immunity" should be removed from the title as they are misleading. One would need to conduct animal studies to show mucosal immunity.*

Response

A revised title is now used based on editor's suggestion: "A cross-reactive human IgA monoclonal antibody blocks SARS-CoV-2 Spike-ACE2 interaction."

Comment 2. *A sequence alignment of the CDRs should also be included in the supplement, not just the framework regions.*

Response

A full sequence alignment including CDRs is now included in the Supplementary Figure 4.

Comment 3. *Even with the improved structural modeling the atomic representations in Figs. 2 and 3 of the mAb362 bound to RBD should be relegated to the supplement to de-emphasize the confidence in the model. The new mutagenesis data is highly supportive of the antibody overlapping the ACE2 binding site but the accuracy of the model really needs to be validated with an explicit structure.*

Response

Overall, we believe the structural models add critical insights to the manuscript and helps explain our rationale for the mutational scans and results. We have edited the figure captions to further reinforce that the figures are molecular models. We agree old-Figure 2d is bit too detailed in the prediction of which part of the antibody is likely interacting with the RBD and have moved this to the Supplementary Data section as Supplementary Figure 3. We feel the rest of the panels in Figure 2 and 3 are key to the understanding of the likely molecular mechanism of MAb362, by validating these through mutagenesis. We also believe Figure 3 is key to show how the predicted binding mode from our mutation guided modeling compares to other reported antibodies and therefore would like to keep it remain in the main body of the manuscript – unless the editor explicitly disagrees. We have in fact tried extensively to co-crystallize this complex, but we were not successful in obtaining diffraction quality crystals.

Reviewer #2 (Remarks to the Author):

Comment 1. The authors have made a substantial effort to address all of the points raised previously, and the resulting revised manuscript is greatly improved. In particular, the addition of live virus neutralisation experiments has strengthened their conclusions. I am mostly satisfied with the responses to my specific concerns (many of which were shared by reviewer 1). I would urge the authors to take more care when drafting rebuttal letters in future as they refer to the wrong supplementary figures on several occasions.

Response: Thank you so much for pointing this out and we apologize for the confusion. We will ensure that this does not happen again.

Comment 2. The construct and purification strategy used to obtain the spike ectodomains is appropriate and I am satisfied by the ELISA results. However, for the sake of being thorough, I'd like to point out that proteins containing multimerization domains, such as T4 fibritin, can often elute at the expected molecular weight by SEC even if they are not adopting the desired 'intact' quaternary structure.

Response:

Thank you for the comment. We completely agree that the expected molecular size may not be a direct indication of intact quaternary structure. We used the construct and purification protocol provided by Dr. Corbett's, who used the same construct and method to obtain a CryEM data of trimer for their recent publication (Wrapp, D. *et al.* Cryo-EM structure of the 2019-nCoV spike in the prefusion conformation. *Science* **367**, 1260-1263, 2020). Though we are confident of the quaternary structure, it is in our future plan to confirm the quaternary structure of our spike trimer.

Comment 3. Given the degree of sequence similarity to 80R, and the mutagenesis data, I'd say the authors homology model is in the right ballpark. By toning down their interpretation of the epitope, the results are less overstated. That being said, X-ray crystallography and/or cryo-EM data would significantly improve the paper, but I appreciate this is outside the scope of a reasonable reviewer request. Overall, given the important nature of the work, I believe the manuscript is sufficiently improved to warrant publication.

Response:

We agree with the reviewer that an experimental structure is outside the scope of the current work. We have been actively pursuing crystallography but have been unsuccessful in obtaining diffraction quality crystals. We also feel that the homology model is overall good and agrees with experimental mutational data, so adds to the overall insights of the manuscript. We appreciate the reviewer's reasonable comment. Thank you so much for your support for the acceptance!